# SemFlow: Binding Semantic Segmentation and Image Synthesis via Rectified Flow

**Chaoyang Wang**[1]    **Xiangtai Li**[1]    **Lu Qi**[2]    **Henghui Ding**[3]
**Yunhai Tong**[1]    **Ming-Hsuan Yang**[2]
[1]School of Intelligence Science and Technology, Peking University
[2]UC, Merced    [3]Institute of Big Data, Fudan University
Project page: https://wang-chaoyang.github.io/project/semflow
cywang@stu.pku.edu.cn, qqlu1992@gmail.com, xiangtai94@gmail.com

## Abstract

Semantic segmentation and semantic image synthesis are two representative tasks in visual perception and generation. While existing methods consider them as two distinct tasks, we propose a unified framework (SemFlow) and model them as a pair of reverse problems. Specifically, motivated by rectified flow theory, we train an ordinary differential equation (ODE) model to transport between the distributions of real images and semantic masks. As the training object is symmetric, samples belonging to the two distributions, images and semantic masks, can be effortlessly transferred reversibly. For semantic segmentation, our approach solves the contradiction between the randomness of diffusion outputs and the uniqueness of segmentation results. For image synthesis, we propose a finite perturbation approach to enhance the diversity of generated results without changing the semantic categories. Experiments show that our SemFlow achieves competitive results on semantic segmentation and semantic image synthesis tasks. We hope this simple framework will motivate people to rethink the unification of low-level and high-level vision.

## 1  Introduction

Understanding semantic content and creating images from semantic conditions are fundamental research topics in computer vision. Semantic segmentation [45; 7; 59; 85; 11; 10] and image synthesis [8; 32; 28; 72; 90] are two representative dense prediction tasks and inspires various downstream applications, including autonomous driving [5] and medical image analysis [89]. The former aims to assign a category label to each pixel in the image, while the latter aims to generate realistic images given semantic layouts.

Although semantic segmentation and image synthesis constitute a pair of reverse problems, existing works typically solve them using two distinct methodologies. On the one hand, segmentation models mostly follow the spirit of discriminative models. Specifically, a pre-trained backbone is employed to extract multi-scale features, and then task-specific decoders are used for dense prediction. On the other hand, semantic image synthesis frameworks are mainly built upon generative adversarial networks (GAN) [19; 52; 57; 62] or diffusion models (DM) [63; 64; 25]. GAN-based methods [90; 72; 28] typically take semantic layouts as inputs and adopt a discriminator for adversarial training. Meanwhile, DM-based methods [36] generate from noise with semantic layouts functioning as control signals.

An intuitive solution is to represent the segmentation mask as a colormap and model it as a conditional image generation task [56]. However, there are several implementation challenges. Overall, most of

---

Corresponding author: Lu Qi and Xiangtai Li.

38th Conference on Neural Information Processing Systems (NeurIPS 2024).

Forward Flow ODE (Semantic Segmentation)

$z_0 \xrightarrow{\quad\quad\quad\quad\quad dz_t = v(z_t, t)dt \quad\quad\quad\quad\quad} \tilde{z}_1$

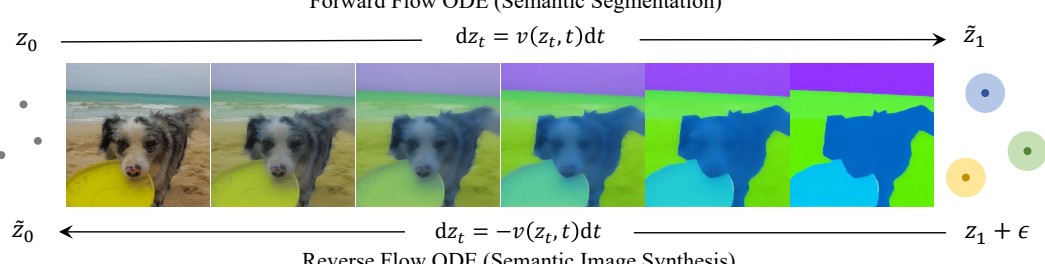

$\tilde{z}_0 \xleftarrow{\quad\quad\quad\quad\quad dz_t = -v(z_t, t)dt \quad\quad\quad\quad\quad} z_1 + \epsilon$

Reverse Flow ODE (Semantic Image Synthesis)

Figure 1: **Rectified flow bridges semantic segmentation (SS) and semantic image synthesis (SIS).** SS and SIS are modeled as a pair of transportation problems between the distributions of images and masks. They share the same ODE and only differ in the direction of the velocity field. We propose a finite perturbation operation on the mask to enable multi-modal generation without changing the semantic labels. *Grey dots* represent data samples. *Colored dots* represent semantic centroids, also known as anchors in Eq. 7. *Colored bubbles* represent the scale of perturbation.

the discriminative segmentation models do not apply to this problem due to their irreversibility. For generative adversarial networks, their generators are typically unidirectional. Jointly training multiple unidirectional models to achieve bidirectional generation [90] is not the concern of this paper. Latent diffusion models (LDMs) have recently demonstrated great potential in generative tasks. Beyond generative tasks, several works [56; 69] attempt to apply diffusion models for segmentation, with images functioning as conditions, but their applicable tasks are limited to be class-agnostic. There are three main problems for existing LDM-based segmentation frameworks: 1) The contradiction between the randomness of generation outputs and the certainty of segmentation labels. 2) The huge cost brought by multiple inference steps. 3) The irreversibility between semantic masks and images.

In this paper, we use rectified flow [40; 42; 17] to enable LDM as a unified framework for semantic segmentation and semantic image synthesis. Our key idea is summarized in Fig. 1. Starting from an LDM [58] framework, we solve the above problems with three methodologies. First, we redefine the mapping function to address the randomness. Previous works [56] aim to learn the mapping from the joint distribution of Gaussian noise and images to the segmentation masks. We argue that Gaussian noise in this mapping function is redundant and negatively affects the determinism of semantic segmentation results. Instead, our model *directly* learns the mapping from images to masks. Secondly, we make the mapping reversible via rectified flow. Rectified flow is an ordinary differential equation (ODE) framework with a time-symmetric training object. This feature allows the model to be trained once to obtain bi-directional transmission capabilities and can be solved numerically using simple ODE solvers such as Euler. Finally, we propose a finite perturbation method to enable multi-modal image synthesis, as the mapping is one-to-one in semantic segmentation but one-to-many in semantic synthesis. We add amplitude-limited noise, enabling masks to be sampled from a collection of semantic-invariant distributions rather than a fixed value, and further improving the quality of synthesized results. Moreover, our model needs fewer inference steps than traditional diffusion models because the transport trajectory is straight, significantly reducing the gap between LDM and traditional discriminative models in segmentation.

The main contributions are as follows: 1) We introduce SemFlow, a new unified framework that binds semantic segmentation and image synthesis with rectified flow, effectively leveraging the length of the generative models. 2) We propose specialized designs of SemFlow, including pseudo mask modeling, bi-directional training of segmentation and generation, and a finite perturbation strategy. 3) We validate SemFlow on several popular benchmarks. For semantic segmentation, SemFlow dramatically narrows the gap between diffusion models and discriminative models in terms of accuracy and inference speed with a more elegant framework. Meanwhile, SemFlow also performs decently in semantic image synthesis tasks.

## 2  Related Work

**Diffusion Models and Rectified Flow.** Diffusion models [25; 63; 64] have shown impressive results in the field of generation, such as image generation [58; 55; 83; 27; 13; 54], video generation [80],

image editing [4; 46; 60; 51; 23], image super resolution [61; 26] and point cloud [47; 50; 88; 82; 48]. Most of these methods are based on stochastic differential equations (SDEs) and need multiple steps for generation. Recently, some works [40; 39; 38; 1; 22] propose to model with probability flow ordinary differential equations (ODEs) to reduce the inference steps. Specifically, rectified flow [40; 42] defines the forward process as straight paths and uses reflow to minimize the sampling steps to one. Although rectified flow has demonstrated decent results on image generation and image transfer tasks, they are limited to low-level vision and lack an exploration of the unification of segmentation and generation tasks.

**Semantic Segmentation** is one of the core tasks in visual perception, aiming to assign each pixel of the given image a semantic category. Previous semantic segmentation approaches are typically built upon discriminative modeling, consisting of a strong backbone [21; 16; 43; 44; 33] for feature extraction and a task-specific decoder head [7; 59; 85; 11; 10; 87; 65; 81; 15; 14] for mask prediction. Recently, some works [34; 86; 18; 77; 75; 3; 70; 9; 20; 2; 29; 76; 71] exploit diffusion models for segmentation. They typically follow the spirit of discriminative models and employ the diffusion model as a feature extractor. Although UniGS [56] and LDMSeg [69] attempt to use a plain Stable Diffusion framework for segmentation, their applicable tasks are limited to be class-agnostic. In practice, reconciling the stochastic outputs of diffusion models with the deterministic results of semantic segmentation is difficult. We rethink this problem and re-model it with rectified flow.

**Semantic Image Synthesis** is the reverse problem of semantic segmentation, aiming to generate realistic images given semantic layouts [8; 53; 41; 74; 91; 66; 67; 79; 32; 35; 49]. Several studies have delved into semantic synthesis through two methodologies. One methodology [90; 28; 72; 6; 68] is based on GAN and trained with adversarial loss and reconstruction loss. However, some of these methods can only generate unimodal outputs. Although some methods [67; 92] have been developed to address the diversity issues, GAN-based frameworks typically suffer from unstable training and need careful parameter tuning. Another methodology [73] employs diffusion models and regards it as a conditional image generation task, where semantic masks function as control signals. Some works further add a greater variety of control signals to enhance the consistency of synthesized results, like textual prompts [78] and bounding box [36]. Despite these methods achieves good synthetic results, their architecture is usually asymmetric, and the generator is usually unidirectional. This hinders the exploration of the unification of semantic segmentation and image synthesis.

## 3 Method

In this section, we first review diffusion models and the differential equations of diffusion-based segmentation models (DSMs) and then analyze the disadvantages of existing approaches. Afterward, we propose our SemFlow, which is inspired by rectified flow theory. It solves the randomness problem in the existing DSM and unifies semantic segmentation and image synthesis with *one* model. We will elaborate on each phase of our method.

### 3.1 Diffusion Model and Segmentation Modeling.

Diffusion models are a class of likelihood-based models that define a Markov chain of forward and backward processes. In the forward process, the noise is gradually added to the real data $x_0$ to form the noisy data $x_t$:

$$q(x_t|x_0) = \mathcal{N}(x_t; \sqrt{\alpha_t}x_0, (1 - \alpha_t)I), \quad t \in [0..T], \tag{1}$$

where $\alpha_t$ is functions of $t$. During training, the model $\epsilon_\theta$ learns to estimate the noise by minimizing the objective with L2 loss:

$$\mathcal{L} = \mathbb{E}_{x_0 \sim q(x_0), \epsilon \sim \mathcal{N}(0,I), t} \left[ \left|\left| \epsilon_\theta(\sqrt{\alpha_t}x_0 + \sqrt{1 - \alpha_t}\epsilon_t) - \epsilon_t \right|\right|_2^2 \right], \tag{2}$$

In the backward process, the model starts from Gaussian noise and gradually denoises to generate realistic data [63] via,

$$x_{t-1} = \sqrt{\alpha_{t-1}} \left( \frac{x_t - \sqrt{1 - \alpha_t}\epsilon_\theta(x_t, t)}{\sqrt{\alpha_t}} \right) + \sqrt{1 - \alpha_{t-1} - \sigma_t^2}\epsilon_\theta(x_t, t) + \sigma_t\epsilon_t. \tag{3}$$

Luckily, Eq. 3 provides a unified solution for DDPM and DDIM, where the former belongs to an SDE modeling and the latter is an ODE modeling. We parameterized it as

$$x_0 = f(x_T, \sigma), \tag{4}$$

where $\sigma$ controls the way of modeling:

$$\sigma_t = \begin{cases} \sqrt{(1-\alpha_{t-1})(1-\alpha_t)}\sqrt{1-\alpha_t/\alpha_{t-1}} & \text{DDPM} \\ 0 & \text{DDIM} \end{cases} \tag{5}$$

In the image generation task, $x_0$ represents the real images, and $x_T$ represents the Gaussian noise. We extend the boundaries of Eq. 4 and apply it to semantic segmentation problems. An intuitive way is to encode the segmentation mask into colormaps and model segmentation as a conditional image generation task. Given image $I$, we rewrite the projection $f$ into a formulation of conditional generation,

$$x_0 = f(x_T, \sigma, I). \tag{6}$$

Eq. 6 formulates the existing diffusion-based segmentation models with generative modeling. However, this modeling approach suffers from the following problems: **1)** The randomness of initial noise and the determinism of the segmentation mask are contradictory. **2)** From a transmission point of view, the transfer from noise to the segmentation mask does not conform to the paradigm of semantic segmentation task, where noise is redundant. **3)** The images in this approach only serve as the condition, which is non-causal in reverse transfer. Thus, the reversed transfer is a separate problem.

### 3.2 Unify Segmentation and Synthesis with Rectified Flow

**Task-agnostic Framework.** We employ the *standard Stable Diffusion* [58] (SD) framework for our task without any task-specific decoder head or well-designed text prompts. To this end, we design the network architecture using the following three steps. First, we convert the semantic segmentation masks to 3-channel pseudo mask $M = (m_0, m_1, m_2)$ to align with the images $I$. Assume that the valid region $[0, 255]$ (for real images) is divided into $k$ parts with a spacing of $s$, the pseudo mask corresponding to category index $c$ is formulated as:

$$m_0' = \lfloor c/k^2 \rfloor, \; m_1' = \lfloor (c - m_0' * k^2)/k \rfloor, \; m_2' = c - m_0' * k^2 - m_1' * k, \; M = s * (m_0', m_1', m_2'), \tag{7}$$

where $s * (k - 1) < 255$ and $\lfloor \cdot \rfloor$ means the floor operator. The $(m_0^i, m_1^i, m_2^i)$ are called anchors.

After the transformation, we adopt a VAE encoder $\mathcal{E}$ to compress the images and pseudo masks into the latent space. The corresponding VAE decoder $\mathcal{D}$ restores the latent variables to the pixel space.

$$z_0 = \mathcal{E}(I), \quad z_1 = \mathcal{E}(M), \quad \tilde{I} = \mathcal{D}(\tilde{z}_0), \quad \tilde{M} = \mathcal{D}(\tilde{z}_1), \tag{8}$$

where $\tilde{z}$ means the output of the UNet.

***Discussion.*** Although previous works [69] argue that segmentation masks are lower in entropy and specifically re-train a lighter network, we still adopt the off-the-shelf VAE from SD. The reasons are as follows: 1) The number of parameters in VAE is negligible compared to the UNet (84M vs. 860M). 2) The VAE specifically trained for segmentation masks can not be applied to images, thus destroying the bi-directional transportation. 3) Specific training creates spatial priors, such as clustering, which hinders our exploration of LDM's segmentation capability itself.

Note that we do not use image captions or image features as prompts. As our model unifies semantic segmentation and image synthesis with *one* model, the usage of captions contradicts the definition of semantic segmentation task while the features are non-causal for image synthesis. Thus, in this work, we set the prompt as empty.

**Bi-directional Training and Inference.** Contrary to the conventional approach, we propose modeling the segmentation task with rectified flow. It is an ODE framework that aims to learn the mapping between two distributions through straight trajectories.

Denote $\pi_0$ and $\pi_1$ represent the distribution of the latent variables of images and masks, respectively. Denote $z_0 \sim \pi_0$ and $z_1 \sim \pi_1$, the trajectory from $z_0$ to $z_1$ can be formulated as $z_t = \varphi_t(z_0, z_1)$:

$$\frac{\mathrm{d}z_t}{\mathrm{d}t} = \frac{\partial \varphi_t(z_0, z_1)}{\partial t}. \tag{9}$$

When $\varphi$ is the trajectory of rectified flow [40], $z_t$ can be reformulated as the linear interpolation process between $z_0$ and $z_1$ as $z_t = (1-t)z_0 + tz_1$. We aim to learn the velocity field using neural

networks $v_\theta(z_t, t)$ and solve it with optimization methods,

$$\mathcal{L} = \int_0^1 \mathbb{E}_{(z_0, z_1) \sim \gamma} \left[ \left|\left| v_\theta(z_t, t) - \frac{\partial \varphi_t(z_0, z_1)}{\partial t} \right|\right|^2 \right] \mathrm{d}t, \tag{10}$$

$$= \int_0^1 \mathbb{E}_{(z_0, z_1) \sim \gamma} \left[ ||v_\theta(z_t, t) - (z_1 - z_0)||^2 \right] \mathrm{d}t,$$

where $\gamma$ indicates the coupling of images and their corresponding pseudo masks.

Eq. 10 is our training loss. Upon training completed, the transfer from $z_0$ to $z_1$ can be described via an ODE:

$$\frac{\mathrm{d}z_t}{\mathrm{d}t} = v_\theta(z_t, t), \quad t \in [0, 1]. \tag{11}$$

So far, we have constructed a mapping from the distribution of images to that of masks. Compared with DSMs in Eq. 6, our approach avoids the interference of randomness, enabling the application of diffusion models to semantic segmentation tasks.

Moreover, Eq. 10 has a time-symmetric form, which results in an equivalent problem by exchanging $z_0$ and $z_1$ and flipping the sign of $v_\theta$. Interestingly, the transportation problem from $\pi_1$ to $\pi_0$ indicates the semantic image synthesis task. This means that semantic segmentation and semantic image synthesis essentially become a pair of mutually reverse problems that share the same ODE (Eq. 11) and have solutions with opposite signs.

In the inference stage, we can obtain the approximate results by numerical solvers like the forward Euler method, which can be formulated as follows,

Semantic segmentation is regarded as the transportation from $z_0$ to $z_1$, which we call *forward flow*. The ODE is Eq. 11 and the numerical solution is,

$$z_{t+\frac{1}{N}} = z_t + \frac{1}{N} v_\theta(z_t, t), \quad t \in \{0, 1, ..., N-1\}/N. \tag{12}$$

After $\tilde{M}$ is restored with Eq. 8, we calculate the L2 distance between $\tilde{M}$ and anchors and obtain the segmentation mask.

Correspondingly, the semantic image synthesis task is considered to transfer in a reverse direction, which we call the *reverse flow*. Specifically, this transfer can be described as $\frac{\mathrm{d}z_t}{\mathrm{d}t} = -v_\theta(z_t, t)$, and the solution is,

$$z_{t-\frac{1}{N}} = z_t - \frac{1}{N} v_\theta(z_t, t), \quad t \in \{N, N-1, ..., 1\}/N. \tag{13}$$

### 3.3 Finite Perturbation

In Eq. 10, the model is configured to learn the one-to-one mapping between images and masks. However, assigning a fixed mask to each image brings about several problems. First, we find that constant $z_1$ in Eq. 13 hinders the multi-modal generation for the image synthesis task. Moreover, the pseudo masks are low in entropy. We hypothesize that the low-entropy distribution of masks hinders the training process and may finally spoil the quality of synthesis results.

To this end, we propose to add finite perturbation on the pseudo masks. Specifically, given pseudo masks $M$ which has a spacing of $s$, we add a noise with a limited amplitude on $M$,

$$M' = M + \epsilon, \quad \epsilon \sim \mathrm{U}(-\beta, \beta), \tag{14}$$

where U is a uniform distribution. We set $0 < \beta < s/2$ to ensure that the semantic label of each pixel does not change. Therefore, we propose to replace $z_1$ with $z_1' = \mathcal{E}(M')$ in Eq. 10 and Eq. 13. We show the effectiveness of this design in Sec. 4.3.

## 4 Experiments

### 4.1 Experimental Settings

**Dataset and Metrics.** We study SemFlow using three popular datasets: COCO-Stuff [37], CelebAMask-HQ [30], and Cityscapes [12]. They contain 171, 19, and 19 categories, respectively. For semantic segmentation, we evaluate with mean intersection over union (mIoU). For

Table 1: **Semantic segmentation results on COCO-Stuff dataset**. **SS** and **SIS** represents semantic segmentation and semantic image synthesis, respectively. **Sampler-N** means the usage of a specific sampler with N inference steps.

| Method | Task | | Sampler | COCO-Stuff | CelebAMask-HQ | |
| | SS | SIS | | mIoU (SS) | FID (SIS) | LPIPS (SIS) |
| --- | --- | --- | --- | --- | --- | --- |
| MaskFormer [11] | ✓ | | - | 41.9 | - | - |
| DSM [58] | ✓ | | DDIM-200 | 16.1 | - | - |
| DSM [58] | ✓ | | DDPM-200 | 20.2 | - | - |
| pix2pixHD [72] | | ✓ | - | - | 54.7 | 0.529 |
| SPADE [53] | | ✓ | - | - | 42.2 | 0.487 |
| SC-GAN [74] | | ✓ | - | - | 19.2 | 0.395 |
| BBDM [31] | | ✓ | BBDM-200 | - | 21.4 | 0.370 |
| SemFlow (ours) | ✓ | ✓ | Euler-25 | 38.6 | 32.6 | 0.393 |

semantic image synthesis, we assess with the Fréchet inception distance (FID) [24] and learned perceptual image patch similarity (LPIPS) [84].

**Implementation Details.** The SemFlow model is built upon Stable Diffusion UNet and initialized with weights from the pre-trained SD 1.5. Images and semantic masks in COCO-Stuff, CelebAMask-HQ, and Cityscapes are resized and cropped into $512 \times 512$, $512 \times 512$, and $512 \times 1024$, respectively. We use the off-the-shelf VAE of the corresponding Stable Diffusion model. The spacing $s$ is set as 50, and the division coefficient $k$ is 6. The amplitude of perturbation is 6. Note that there is no need to train two models for segmentation and synthesis separately since the optimization target in Eq. 10 is time-symmetric.

**Baselines.** We seek to unify semantic segmentation (SS) and semantic image synthesis (SIS) into a pair of reverse problems and train *one* model for a solution. As few previous works achieve this goal, we thus take different baselines for SS and SIS, respectively. For SS, we design two baseline models, DSMs [58], which follow the principle of diffusion-based conditional generation modeling in Sec. 3.1. The network structure and hyperparameters of DSM follow SemFlow, except that the inputs of UNet are eight channels. Specifically, the noise and images are concatenated in the channel dimension. DSM-DDIM and DSM-DDPM differ in differential equation modeling (ODE vs. SDE). For SIS, we take pix2pixHD [72], SPADE [53], SC-GAN [74], BBDM [31] and CycleGAN [90] as the baselines.

## 4.2 Main Results

**Semantic Segmentation.** Fig. 2 shows the qualitative comparison between SemFlow and DSMs. Our SemFlow demonstrates satisfactory performance in a range of scenarios and exhibits high accuracy in classifying the semantic labels of targets. In contrast, DSM fails in semantic segmentation tasks, regardless of SDE or ODE modeling. First, DSM is inferior to SemFlow in discriminating different semantic categories. Moreover, the outputs of DSM-DDIM and DSM-DDPM change dramatically with different random seeds. As discussed in Sec. 3.1, DSM is susceptible to the randomness inherent in diffusion models, making it unable to produce deterministic results.

Tab. 1 shows the quantitative results on the COCO-Stuff dataset. We compare SemFlow with two variants of DSM and MaskFormer, which is a classical discriminative segmentation model. Regarding accuracy, our SemFlow achieves 38.6 mIoU and outperforms DSMs by a remarkable margin. Moreover, SemFlow only uses a simple sampler, forward Euler method, and fewer inference steps than DSMs. Further reducing the inference steps brings about faster generation but witnesses a slight drop in performance, which is analyzed in Sec. 4.3.

**Semantic Image Synthesis.** We compare our approach with other specialist models on semantic image synthesis tasks in Tab. 1. On CelebAMask-HQ, SemFlow achieves a performance of 32.6 FID and 0.393 LPIPS.

Beyond specialist models, we also qualitatively compare our approach with CycleGAN [90] in Fig. 3 to demonstrate the *overall* performance on SS and SIS tasks. All results of SemFlow are generated with *one* model. In other words, the ODE modeling of SS and SIS share the same velocity

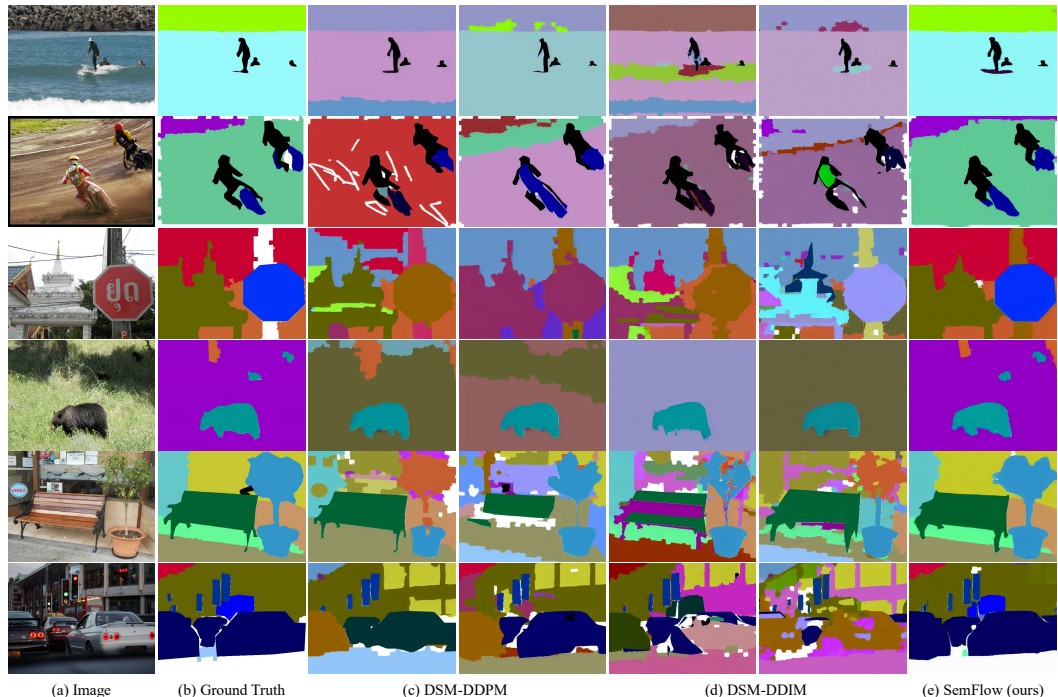

| (a) Image | (b) Ground Truth | (c) DSM-DDPM | (d) DSM-DDIM | (e) SemFlow (ours) |

Figure 2: **Semantic segmentation results on COCO-Stuff dataset.** For the ground truth, each color reflects the value of anchors (Eq. 7), which corresponds to one semantic category, and the color white indicates the ignored regions. The predictions of DSM vary considerably under different random seeds.

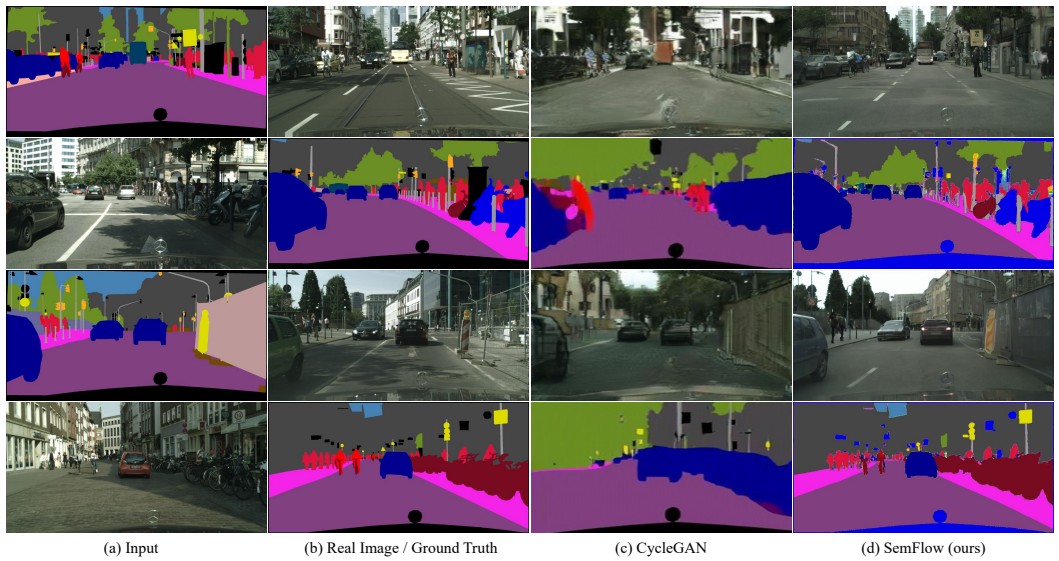

| (a) Input | (b) Real Image / Ground Truth | (c) CycleGAN | (d) SemFlow (ours) |

Figure 3: **Semantic segmentation and semantic image synthesis results on Cityscapes dataset.** The color black in the ground truth indicates the ignored region. The segmentation results of SemFlow are colored following [12].

field and only differ in the sign. The first and third rows show the image synthesis results given semantic layouts, while the second and fourth row shows the segmentation results. Our synthesis and segmentation results are inspiring and significantly outperform CycleGAN. Note that CycleGAN essentially trains two unidirectional generators while we employ only *one* model for the two tasks.

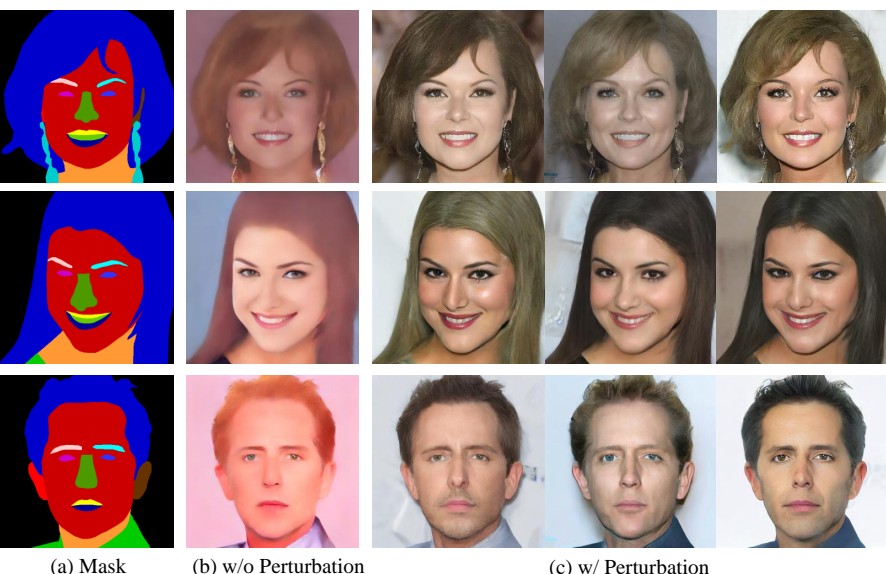

| (a) Mask | (b) w/o Perturbation | (c) w/ Perturbation |

Figure 4: **Semantic image synthesis results on CelebAMask-HQ dataset.** Semantic masks are colored to show different semantic components. SemFlow w/ Perturbation indicates the finite perturbation operation in Eq. 14.

**Discussion.** Although the performance gap exists compared with specialist models, our approach solves the two major issues of diffusion-based segmentation approaches: 1) Randomness of outputs and 2) Large inference steps. We believe it significantly reduces the gap between discriminative models and diffusion models.

### 4.3 Ablation Studies

**Perturbation.** Fig. 4 shows the synthesized results of SemFlow on the CelebAMask-HQ dataset and demonstrates the importance of perturbation operation. First, perturbation enables our approach to generate multi-modal results from one semantic layout using different noises. The synthesized results show diverse appearances, such as skin, hair color, and the background, and exhibit good consistency with the masks. Moreover, we also show the synthesis results of the model without perturbation in Fig. 4. It is witnessed that the model without perturbation is only able to generate uni-modal results, and the synthesized images exhibit a loss of detail and suffer from severe over-smoothing problems.

**Straight Trajectory.** Our SemFlow aims to establish linear trajectories between the distribution of images and the masks, which enables it to sample with fewer steps.

Tab. 2 compares the performance of DSM and SemFlow on segmentation tasks under different inference steps, where DSM sampled with DDPM and DDIM represent SDE and ODE modeling, respectively. Overall, SemFlow outperforms DSM by a remarkable margin regardless of the inference steps. Specifically, the DSMs' performance declines markedly as the number of steps decreases. With 5 steps, the DSM-DDPM achieves a mIoU of 12.3, while the DSM-DDIM achieves 9.5 mIoU. On the contrary, our SemFlow achieves 28.3 even with one step.

Fig. 5 provides semantic image synthesis results with different inference steps. Although fewer steps will result in some loss of detail and make the images appear smoother, the results of one-step synthesis are still competitive. This indicates that the trajectory between the two distributions is straightened. Fig. 6 and Fig. 7 visualizes the latent variable sampled from the trajectory. The transition between $z_0$ and $z_1$ is smooth.

## 5 Conclusion

In this work, we present SemFlow, a framework that employs a diffusion model to integrate semantic segmentation and semantic image synthesis as a pair of reverse problems. To overcome the con-

Table 2: **Semantic segmentation results with different inference steps on COCO-Stuff dataset.** mIoU is used as the metric.

| Method&Sampler | | 1 | 2 | 5 | 10 | 25 | 50 | 100 | 200 |
|---|---|---|---|---|---|---|---|---|---|
| DSM | DDIM | - | 0.1 | 4.0 | 9.5 | 13.6 | 14.9 | 15.7 | 16.1 |
| | DDPM | - | 0.1 | 5.3 | 12.3 | 17.5 | 18.9 | 19.6 | 20.2 |
| SemFlow | Euler | 28.3 | 31.0 | 36.9 | 38.4 | 38.6 | 38.4 | 38.3 | 38.3 |

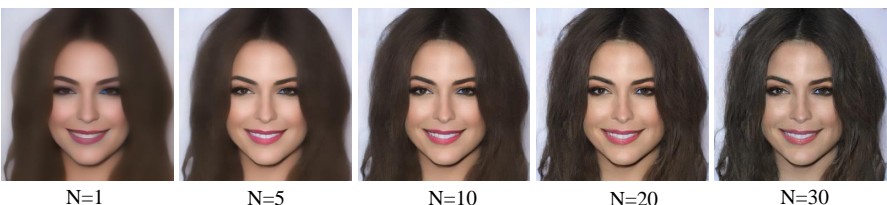

N=1     N=5     N=10     N=20     N=30

Figure 5: **Image synthesis results with different inference steps.** We use the forward Euler method to get numerical solutions. Our approach obtains competitive results even with only one inference step.

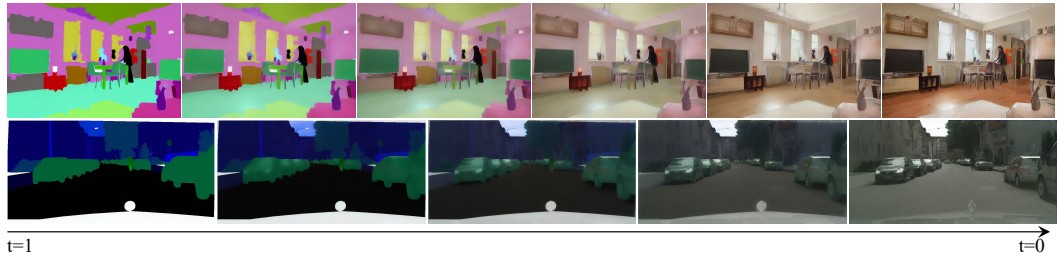

t=1                            t=0

Figure 6: **Visualization of latent variables on the trajectory from $z_1$ to $z_0$ (Semantic image synthesis).** Top row: COCO-Stuff. Bottom row: Cityscapes.

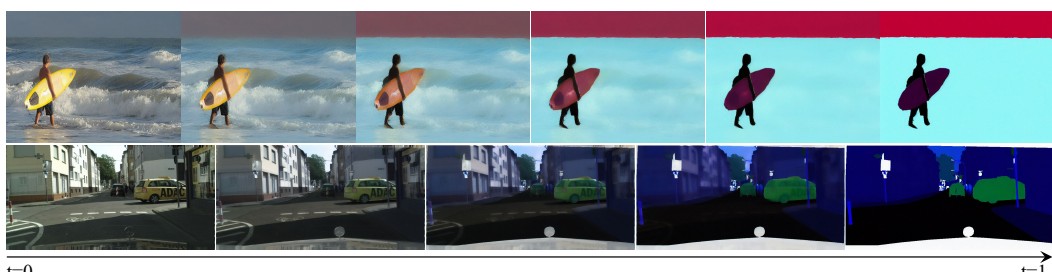

t=0                            t=1

Figure 7: **Visualization of latent variables on the trajectory from $z_0$ to $z_1$ (Semantic segmentation).** Top row: COCO-Stuff. Bottom row: Cityscapes.

tradiction between the randomness of diffusion models and the certainty of semantic segmentation results, we propose to model semantic segmentation as a transport problem between image and mask distributions. We then employ rectified flow to learn the transfer function, which brings the benefits of reversible transportation. We propose a finite perturbation method to enable multi-modal generation, which also greatly improves the quality of synthesized results. With straight trajectory modeling, our model can sample with much fewer steps. Experimental results show that even with a weak sampler, our model still achieves comparable or even better results than specialist models. We hope our research can inspire the findings on unified generative model design for the community.

**Acknowledgement.** This work was supported by the National Key Research and Development Program of China (No. 2023YFC3807600).

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

**Overview.** In this supplementary, we present more results and details:

- **A.** More details on our method.
- **B.** More empirical studies and visual results.
- **C.** Discussion of the limitations, broader impacts and safeguards.

# A    More Details on Method

**Amplitude of Perturbation.** In the main paper, the perturbation follows the uniform distribution $\epsilon \sim \mathrm{U}(-\beta, \beta)$. We set $0 < \beta < s/2$ to prevent the semantic labels from changing. Here is the proof.

Let $M = (m_0, m_1, m_2)$ be the anchor for category $c_1$, $M' = M + \epsilon$ be the anchor with perturbation, and $P = (p_0, p_1, p_2)$ be another anchor for category $c_2, c_1 \neq c_2$. First, it is obvious that $E(M') = E(M) = M$, which means that adding perturbation will not change the estimation of original anchors. Then we only need to prove $||M' - M||_2 < ||M' - P||_2$, which is equivalent to $\sum_{i \in [3]} \epsilon_i^2 < \sum_{i \in [3]} (\epsilon_i + m_i - p_i)^2$, and can be further written as $0 < \sum_{i \in [3]} (m_i - p_i)(m_i - p_i + 2\epsilon_i)$. Since $-s < 2\epsilon_i < s$ and $m_i - p_i$ can only take the values of $\{..., -2s, -s, 0, s, 2s, ...\}$, the two terms must have the same sign or one of them equals 0. The proof is completed.

**Implementation Details.** We train CelebAMask-HQ and Cityscapes with a batch size of 256 with AdamW optimizer for 80K and 8K steps, respectively. The initial learning rate is set as $2 \times 10^{-5}$ and $5 \times 10^{-5}$. Linear learning rate scheduler is adopted. For COCO-Stuff dataset, we use a constant learning rate of $1 \times 10^{-5}$ with a batch size of 128 for 320K steps. The COCO-Stuff indicates the version of COCO-Stuff-164k.

# B    More Empirical Studies and Visual Results

**Sampling from Other Distributions.** Fig. 8 shows the synthesized images which are sampled with different perturbations $\epsilon \sim \mathrm{U}(-\beta', \beta')$. It can be observed that the model can obtain high-quality synthetic results only when the perturbations in the inference stage and the training stage follow the same distribution. Specifically, the synthesized images are dim when $\beta' < \beta$, and overexposed when $\beta' > \beta$.

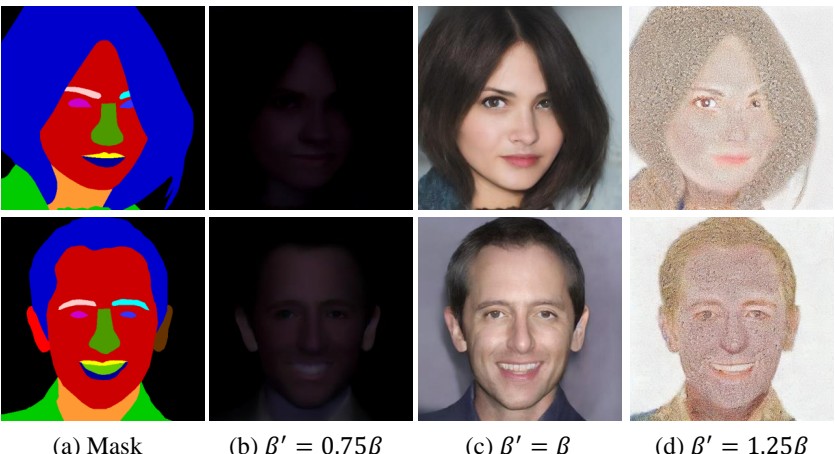

|  (a) Mask  |  (b) $\beta' = 0.75\beta$  |  (c) $\beta' = \beta$  |  (d) $\beta' = 1.25\beta$  |

Figure 8: **Image synthesis results sampled from other distributions.** We sample with different perturbation $\epsilon \sim \mathrm{U}(-\beta', \beta')$. In the training stage, the perturbation follows $\mathrm{U}(-\beta, \beta)$.

**More Visual Results on One-step Generation.** Fig. 9 shows that our model can generate competitive images with only one inference step.

**Selection of ODE solvers.** Fig. 10 provides visualization results on different ODE solvers. A stronger solver has the potential to bring about better results.

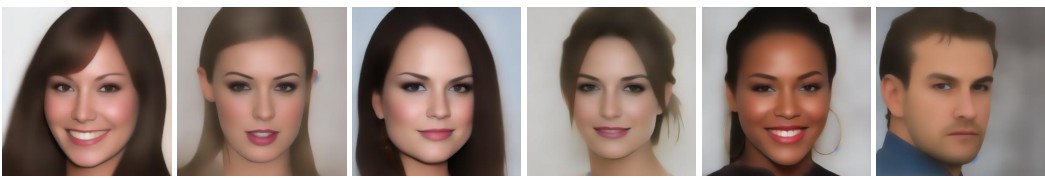

Figure 9: **Synthesized images with one inference step.**

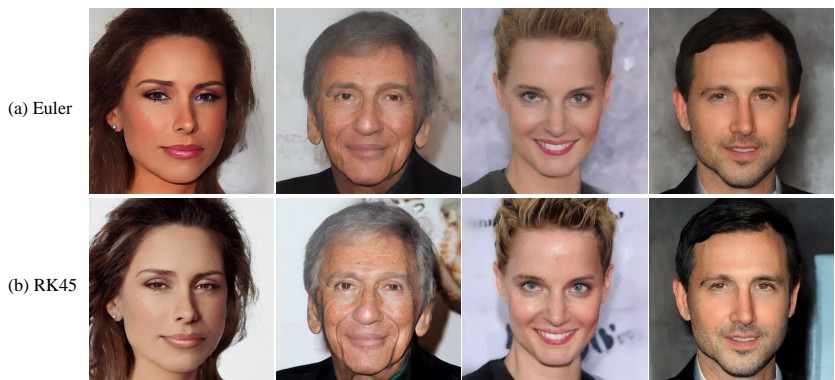

Figure 10: **The influences of ODE solvers.** (a) Euler indicates sampling with euler-25 solver. (b) RK45 indicates the Runge-Kutta method of order 5(4).

**More Visual Results on Cityscapes.** Fig. 11 shows the results of multi-modal image synthesis. The appearance of the synthesized scene varies according to different random seeds. Moreover, our model can learn the laws of shadows and light angles (the third column), demonstrating its great potential.

Fig. 12 shows visual results of semantic segmentation. Our model accurately segments the objects and assigns the correct semantic labels.

## C  Discussion

**Limitations.** Some previous work, such as Stable Diffusion, SDXL show that scaling up the training data is essential to exploit the potential of diffusion models fully. Due to the limited computational resources and the number of image-mask pairs in the existing dataset, we leave it to future work. In addition, although this paper does not focus on the network structure and the design of ODE solvers, we believe there is room for exploration to further improve performance.

**Broader Impacts.** Our SemFlow extends the diffusion model from the generative domain to the segmentation task and narrows the gap with traditional discriminative models. It also functions as a unified framework to bind semantic segmentation and image synthesis. We hope this will motivate people to explore the unification of low-level and high-level vision. In terms of social impact, it will encourage artistic content creation, but at the same time may face the problem of fake content.

**Safeguards.** The content produced by a generative model is highly correlated with its training data. Ensuring fair and clean training data can effectively prevent models from generating harmful content.

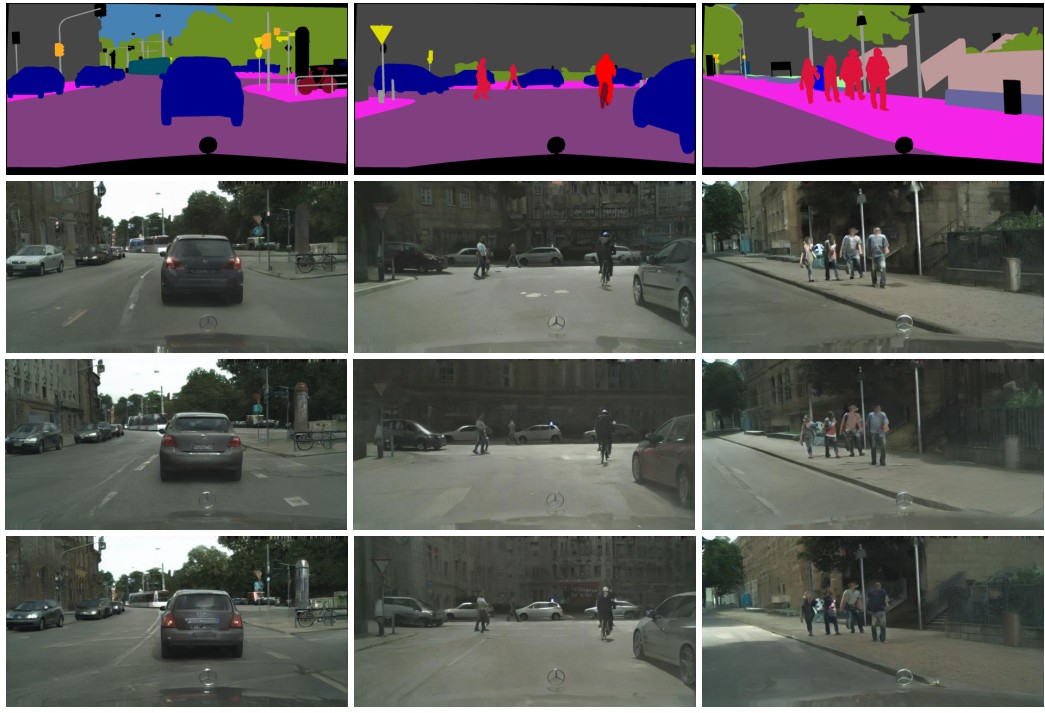

Figure 11: **Image synthesis results on Cityscapes.** We show the results under three random seeds for each semantic mask. The first row: semantic layouts. The second to the fourth row: synthesized results.

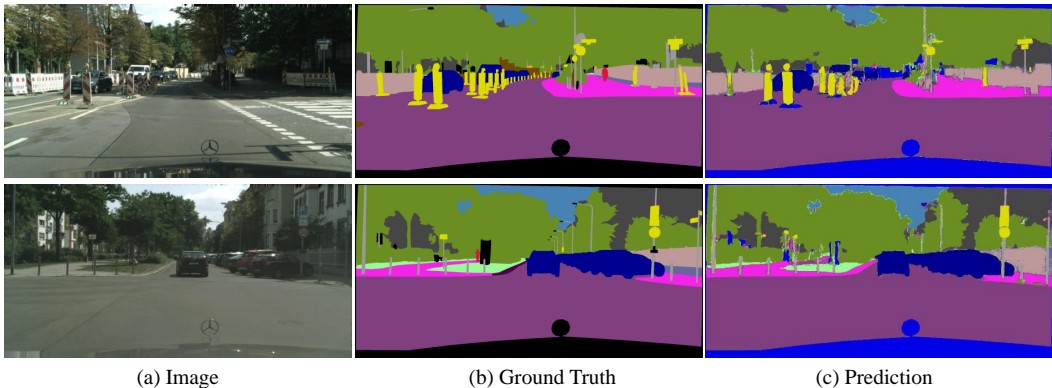

(a) Image                  (b) Ground Truth               (c) Prediction

Figure 12: **Semantic segmentation results on Cityscapes.** The color black in the ground truth indicates the ignored regions.

