# OpenReview forum: "SemFlow: Binding Semantic Segmentation and Image Synthesis via Rectified Flow"
_NeurIPS.cc/2024/Conference — NeurIPS 2024 poster_

### Official Review · Reviewer_eo5D · 2024-07-07

**Soundness:** 4
**Presentation:** 3
**Contribution:** 3
**Rating:** 5
**Confidence:** 3

**Summary:**

This paper presents a work for solving semantic image segmentation and image synthesis simultaneously. This work proposed to use rectified flow and adopt a VAE encoder to compress the images and pseudo masks into the latent space. By comparing the proposed method with previous semantic segmentation models and image synthesis SOTAs, the effectiveness of this method is demonstrated. The proposed method handles the randomness issue for semantic segmentation. Finally, ablation study has been provided.

**Strengths:**

+ This proposed an interesting formulation to bridge semantic segmentation and image synthesis in the stable diffusion framework with rectified flow framework. It is an interesting work and different to previous work on unifying semantic segmentation and image synthesis, which produces segmentation masks and synthesized images at the same time. Differently, this work trains one model to convert between segmentation masks and synthesized images.

+ This work also shows faster generation speed for high quality images.

**Weaknesses:**

- What is the motivation to use Euler sampler. More discussion between Euler and DDIM/DDPM will be more helpful to understand the technical details for most readers.

- For image synthesis task, the proposed method had a clear gap compared with other diffusion models.

**Questions:**

See above.

**Limitations:**

No.

---

> ### Author Rebuttal · Authors · 2024-08-07
>
> Thank you for recognizing our strengths: 1. Our work bridges semantic segmentation and image synthesis with rectified flow framework. 2. Our work shows faster generation speed for high-quality images. We provide more clarifications below.
>
> Q1: The motivation to use the Euler sampler. More discussion between Euler and DDIM/DDPM.
>
> A1: Rectified flow (RF) has one type of modeling, ODE, and diffusion models (DMs) have two types of modeling, SDE and ODE, respectively. We aim to discuss all of these modeling methods. To this end, we use the Euler sampler for SemFlow (RF, ODE) following [1] and DDIM for DSM (DM, ODE) following [2]. Eq.3 provides a unified solution for SDE/ODE modeling of diffusion models, so we adopt DDPM for DSM (DM, SDE). In the context of ODE solvers, DDIM is slightly better as it fully utilizes the semi-linear structure of diffusion ODEs[3].
>
>
> Q2: Performance gap compared with other diffusion models.
>
> A2:
> 1. Our model needs to establish the reversible transport mapping between images and masks. This means it cannot use the transport capability of Stable Diffusion, which is pre-trained on a large image-text dataset and has built stable mappings between noises and images.
> 2. We aim to show that our method can significantly improve the performance and inference speed of diffusion-based segmentation methods. To this end, we follow [1] and only use a simple Euler sampler in segmentation and synthesis tasks. A higher-order sampler has the potential to bring about better results. Here is the comparison on the ade20k dataset,
>
> | Sampler | Euler | RK45 |
> |---|---|---|
> | FID $\downarrow$ | 39.2 | 27.8 |
>
> [1] Liu, Xingchao, et al. "Flow Straight and Fast: Learning to Generate and Transfer Data with Rectified Flow." ICLR. 2023.
>
> [2] Qi, Lu, et al. "Unigs: Unified representation for image generation and segmentation." CVPR. 2024.
>
> [3] Lu, Cheng, et al. "Dpm-solver: A fast ode solver for diffusion probabilistic model sampling in around 10 steps." NeurIPS. 2022.

---

> ### Author Response · Authors · 2024-08-10
> **Please let us know whether we address all the issues**
>
> Dear reviewer,
>
> Thank you for the comments on our paper.
>
> We have submitted the response to your comments. Please let us know if you have additional questions so that we can address them during the discussion period. We hope that you can consider raising the score after we address all the issues.
>
> Thank you

---

> > ### Comment · Reviewer_eo5D · 2024-08-12
> > **Maintain my rating - Borderline accept**
> >
> > Hi
> >
> > After reading all the reviews and authors' feedback, I prefer to maintain my initial rating - borderline accept.
> >
> > From my perspective, this work presents a novel model for jointly modelling image generation and synthesis, which are two reverse tasks. It is clearly different to previous work UniGS and FreeMask, as mentioned by R1.
> >
> > The authors also provide more explanation on the Euler sampler. I think authors need to refine this part much better in the next version, no matter for an accepted paper or a resubmission. Therefore, I insist on my initial rating, even though the performance still needs to improve to compete with other methods.
> >
> > Best
> > R3

---

> > > ### Author Response · Authors · 2024-08-12
> > >
> > > We sincerely appreciate the reviewer's comments and positive feedback. We'll carefully consider your review and make any necessary revisions to improve the quality of our manuscript. Thank you.

---

### Official Review · Reviewer_5PqV · 2024-07-11

**Soundness:** 3
**Presentation:** 3
**Contribution:** 2
**Rating:** 5
**Confidence:** 3

**Summary:**

The paper presents SemFlow, an approach that binds semantic segmentation and semantic image synthesis using an ordinary differential equation (ODE) model. The motivation is to use rectified flow to enable LDM as a unified framework for both tasks. The key contributions include a unified framework that jointly optimizes both tasks, the specialized designs of the frameworks, including pseudo mask modelling, bi-directional training of segmentation and generation, and a finite perturbation strategy. This approach bridges the gap between semantic segmentation and semantic image synthesis, showing good visualization results.

**Strengths:**

1. The unified framework for joint optimization of semantic segmentation and semantic image synthesis is novel.

2. The paper is generally well-written and structured.

**Weaknesses:**

1. Lines 133-140: Why is there a necessity to transform the semantic segmentation masks into 3-channel pseudo-masks utilizing Eq. 7? Could you provide insights on how the formulations for m1 and m2 were derived?

2. Figure 4: During the model inference process (semantic image synthesis task), will noise be added to the mask?

3. Lines 147-155: SemFlow cannot use captions to guide image synthesis, and the authors claim the usage of captions is non-casual for semantic image synthesis, however, there is still some work on it [1,2].

4. There is still a gap in quantitative results compared to the popular model within each task.

[1] Xue, Han, et al. "Freestyle layout-to-image synthesis." Proceedings of the IEEE/CVF Conference on Computer Vision and Pattern Recognition. 2023.

[2] Lv, Zhengyao, et al. "Place: Adaptive layout-semantic fusion for semantic image synthesis." *Proceedings of the IEEE/CVF Conference on Computer Vision and Pattern Recognition*. 2024.

**Questions:**

See Weaknesses

**Limitations:**

Yes

---

> ### Author Rebuttal · Authors · 2024-08-07
>
> Thank you for recognizing our strengths: The unified framework for joint optimization of semantic segmentation and semantic image synthesis is novel. We provide more clarifications below.
>
> Q1: The necessity of 3-channel pseudo-masks and the insights of Eq.7.
>
> A1: The mask needs to be converted into a 3-channel pseudo mask to align with the VAE, which takes as input a 3-channel tensor. Eq.7 is inspired by K-based numbers. In this encoding method, the 3-channel encoding space can represent ${K}^3$ anchors, and the distance between each anchor is greater than $s$. This approach can approximately maximize the utilization of the encoding space.
>
> Q2: Will noise be added to the mask in the semantic image synthesis task?
>
> A2: Yes. The distribution of masks in the inference stage must follow that in the training stage. Also, the noise is a necessity for multi-modal generation. We also conduct ablation experiments in the supplementary Sec. B (Fig.7), which shows that sampling from a different distribution (aka., different noise) brings about significant performance degradation.
>
> Q3: Clarification of "captions" in L147-155.
>
> A3: Sorry for confusion. In L147-150, we wrote, "...we do not use image captions or image features as prompts...the latter is non-casual for image synthesis". The "latter" means "image features". We will revise it to "the image feature is non-causal for image synthesis".
>
> Q4: Performance issues.
>
> A4:
> 1. For semantic segmentation, our model does not use extra feature extractors because they destroy the transport's symmetry.
> 2. From the transport perspective, our model needs to establish the reversible transport mapping between images and masks. However, the mapping of Stable Diffusion is from noises to images. This means that our model cannot use SD's transport capability, so it is a harder task for our model in the image synthesis task.
> 3. Our method can greatly improve diffusion-based segmentation methods' performance and inference speed even with a simple Euler sampler following [1]. A higher-order sampler has the potential to bring about better results. Here is the comparison on the ade20k dataset,
>
> | Sampler | Euler | RK45 |
> |---|---|---|
> | FID $\downarrow$ | 39.2 | 27.8 |
>
> [1] Liu, Xingchao, et al. "Flow Straight and Fast: Learning to Generate and Transfer Data with Rectified Flow". ICLR, 2023.

---

> ### Author Response · Authors · 2024-08-10
> **Please let us know whether we address all the issues**
>
> Dear reviewer,
>
> Thank you for the comments on our paper.
>
> We have submitted the response to your comments. Please let us know if you have additional questions so that we can address them during the discussion period. We hope that you can consider raising the score after we address all the issues.
>
> Thank you

---

### Official Review · Reviewer_3dM2 · 2024-07-12

**Soundness:** 2
**Presentation:** 2
**Contribution:** 2
**Rating:** 4
**Confidence:** 5

**Summary:**

This paper proposed a unified diffusion-based framework for semantic segmentation and semantic image synthesis. The proposed SemFlow applied the existing ordinary differential equation (ODE) model and modified the transport problem setting. Additionally, it incorporates techniques such as perturbation and straight trajectory to enhance model performance. The proposed method was compared with a simple diffusion-based conditional generation modeling and a transformer-based method (Mask2Former) for semantic segmentation, as well as several GAN-based and diffusion-based methods for semantic image synthesis. However, the results did not surpass existing methods, and performance in the semantic segmentation task was even 10% lower in mIoU compared to basic segmentation methods.

**Strengths:**

1. The motivation behind this paper is well though out. It adds value by attempting to unify low-level and high-level vision tasks.
2.The writing style is clear and easy to understand.

**Weaknesses:**

1. The literature review is insufficient and lacks citations of relevant works such as FreeMask, NeurIPS, 2023.
2. Although the motivation is thoughtful, the theoretical and practical value of the proposed method is limited. The application results are not promising, as the semantic segmentation performance is over 10% lower than Mask2Former. The method does not demonstrate advantages over existing training data synthesis methods.
3. The key components of the proposed method are commonly used in generative fields and therefore lack novelty.
4. The experiments are insufficient. It is recommended that the authors include FreeMask and other relevant methods for comparison in the semantic segmentation task, as well as ControlNet or Freestyle generation methods for semantic image synthesis. Additionally, the authors should provide complete quantitative results on all three datasets for both semantic image synthesis and semantic segmentation tasks.
5. The authors claim limited computational resources as a reason to leave training data synthesis methods for future work, yet the experiments were conducted on 8 NVIDIA A100 GPUs, which is more than the resources used in most training data synthesis studies. Also, the computational cost advantages are unclear due to the absence of computation time data.

**Questions:**

Although I appreciate the motivation and exploration in this paper, the results do not convince me that the proposed method is promising and practical for image synthetic and semantic segmentation tasks.

**Limitations:**

Yes

---

> ### Author Rebuttal · Authors · 2024-08-07
>
> Thank you for recognizing the motivation and exploration of our work.
>
> Q1: Lack of citation and comparison with FreeMask in semantic segmentation task.
>
> A1: Thanks for pointing out this. We will cite FreeMask. However, we would like to emphasize that FreeMask is a framework for training data generation, which serves as an augmentation for existing semantic segmentation models, such as Deeplab. FreeMask resorts to synthetic images with an off-the-shelf image synthesis model, FreestyleNet, to enlarge the volume of training data. On the other hand, as stated in the title of our paper, we present a new generative method to bridge semantic segmentation and image synthesis via rectified flow. Therefore, Freemask is totally different from our proposed method in that it has different targets and motivations.
>
> Q2: Novelty issues.
>
> A2: First, we are the first to propose to unify semantic segmentation and semantic image synthesis via rectified flow. Our model first accomplishes **uni-directional training, bi-directional inferencing**. Our approach is simple yet effective without introducing extra components on existing baseline, stable diffusion models. To the best of our knowledge, we are the first to achieve this goal of seamless transformation between images and segmentation masks. Second, our approach greatly improves the performance and inference speed of LDM-based segmentation. Third, we propose finite perturbation to enable multi-modal generation and improve the quality of synthesis results. We think our contributions should not be underestimated. Moreover, other reviewers find our work interesting and novel.
>
> Q3: Performance issues.
>
>
> A3: First, we discuss the performance issues from two perspectives. For segmentation, our model does not rely on strong backbones for feature extraction or task-specific decoder like Mask2Former. For image synthesis, our model needs to establish the transmission from masks to images, rather than from noise to images, which means our model cannot use the transmission capability of Stable Diffusion itself.
>
> Second, we compared SemFlow with Freestyle and ControlNet on ade20k. We use the provided checkpoint from Freestyle and ControlNet for ade20k dataset. Please note that our model does not take text prompts as inputs, so we report two values of Freestyle, respectively, with and without text prompts. Due to the limitation of computational resources, we do not conduct experiments of SemFlow with text prompts.
>
> | Method | Freestyle w/o text | Freestyle w/ text | ControlNet w/ text | ours |
> |---|---|---|---|---|
> | FID $\downarrow$ | 164.5 | 25.0 | 52.1 | 27.8 |
>
> We also use the same model and compare it with MaskFormer on ade20k semantic segmentation task.
>
> | Method | MaskFormer | ours |
> |---|---|---|
> | mIoU $\uparrow$ | 44.5 | 41.0 |
>
> Q4: 8 NVIDIA A100 GPUs are more than the resources used in most training data synthesis studies. Computation time.
>
> A4: We argue that our task is not **training data synthesis**. Our model needs to establish the reversible transport mapping between images and semantic masks. However, the transport of Stable Diffusion (SD) is between noises and images. As a result, our model cannot use the transport capability of SD. The creation of the mapping is difficult. For example, the training of InstaFlow costs around 199 A100 days. We train SemFlow on CelebAMask for around 37 hours.

---

> > ### Comment · Reviewer_3dM2 · 2024-08-13
> >
> > Thank the authors for their response. I have carefully read the responses and the comments from other reviewers. I think my original concerns have been partially addressed. Although the motivation is good, I remain unconvinced that the new framework is more practical or has greater potential in contributing to both image synthesis and semantic segmentation tasks compared to other approaches. The exclusion of text prompts and the inability to leverage existing foundation models reduce the flexibility and practicality of this framework. Additionally, it is unclear how the authors conducted the experiments on Freestyle and ControlNet. It seems that the results for ControlNet are not optimal. Generally, it is not supposed to perform such worse than Freestyle. For these reasons, I am inclined to increase my original score to borderline reject, which is the highest score I can give.

---

> > > ### Author Response · Authors · 2024-08-13
> > >
> > > Thanks for your comments.
> > >
> > > Issue 1: This framework seems unpractical and lacks potential.
> > >
> > > A1: First, our method builds a deterministic transport from images to masks, which solves the stochasticity problem in existing diffusion-based segmentation models (Fig.2, Sec.3.1). Second, our method significantly reduces the sampling steps of diffusion-based segmentation methods through straight trajectory modeling (Tab.3). Third, our framework models image segmentation and generation as a pair of reversible problems. We believe it is helpful for the community and has the potential to achieve better performance when scaled up, as discussed in future works. As for research potential discussion, recently, lots of works use generative model for segmentation. We acknowledge that existing diffusion pipeline for segmentation cannot compete with traditional segmentation models. However, it is also one of solution towards an unified model and should be encouraged considering the diversity of research.
> > >
> > >
> > > Issue 2: The details of experiments on Freestyle and ControlNet.
> > >
> > > A2: For Freestyle, we use the official layout-to-image synthesis script for ADE task. This official script formulates the text prompts as "[class 1] [class 2] ..." where "class n" is the name of the specified category. The checkpoint we use is from the Freestyle official repository.
> > > For ControlNet, we adopt the official checkpoint named "Controlnet - v1.1 - seg Version", which is designed for ADE dataset. We use the provided scripts in README and only replace the text prompts from a complete sentence with a string of category names, which is **in the same way as Freestyle**. We observed that the synthesized results of ControlNet and Freestyle are significantly different in domains, including style, texture, material, etc. This difference is also observed in Freestyle's visualization. The samples of Freestyle are much closer to those of the realistic ADE dataset.

---

> ### Author Response · Authors · 2024-08-10
> **Please let us know whether we address all the issues**
>
> Dear reviewer,
>
> Thank you for the comments on our paper.
>
> We have submitted the response to your comments. Please let us know if you have additional questions so that we can address them during the discussion period. We hope that you can consider raising the score after we address all the issues.
>
> Thank you

---

### Author Rebuttal · Authors · 2024-08-07

## Global Response

We thank all the reviewers for their insightful reviews. We first summarize the strengths of our paper that the reviewers recognized.
1. The unified framework of bridging semantic segmentation and image synthesis via rectified flow is interesting and novel.
2. This work shows faster generation speed for high-quality images.
3. This work adds value by attempting to unify low-level and high-level vision tasks.

Next, we aim to re-elaborate our contributions and address common concerns raised by reviewers.

**Contributions**

This work proposes a unified framework that models semantic segmentation and image synthesis as a pair of reverse problems. We are the first to propose symmetric modeling of segmentation and generation to accomplish **uni-directional training, bi-directional inferencing**.
For semantic segmentation, our approach solves the contradiction between the randomness of diffusion outputs and the uniqueness of segmentation results. It also greatly improves the accuracy and inference speed of diffusion-based segmentation methods.
For image synthesis, we propose a finite perturbation approach to enable multi-modal generation and improve the quality of synthesis results.

**Performance Gap**

Our model aims to establish a reversible transport mapping between the distribution of images and masks. First, for segmentation tasks, we do not use specifically designed backbones for feature extraction (e.g., ResNet, ViT) because they destroy the symmetry of the transport.
Second, Stable Diffusion is pre-trained on large text-image datasets and thus has a strong capability to transport from noises to masks. As a result, diffusion-based methods like Freestyle are easier to obtain good results than to re-establish the mapping.
Finally, we only use a simple Euler sampler to compare fairly with DDIM, a 1-order ODE solver commonly used in diffusion-based segmentation like UniGS[1]. A higher-order sampler has the potential to obtain a more precise estimation of the trajectory and achieve better results.

[1] Qi, Lu, et al. "Unigs: Unified representation for image generation and segmentation." CVPR. 2024.

---

### Decision · Program_Chairs · 2024-09-25

**Decision:**

Accept (poster)

**Comment:**

This paper proposes a new unified framework that binds semantic segmentation and image synthesis with rectified flow as a pair of reverse problems, trained with an ordinary differential equation (ODE) model to transport between the distributions of real images and semantic masks, using techniques including pseudo mask modeling, bi-directional training of segmentation and generation, and a finite perturbation strategy.

It has received 1x borderline reject and 2x borderline accepts.  Authors have provided an extensive rebuttal which has partially addressed most concerns.  Reviewers generally like the motivation to unify two different tasks in a reversible process,  with "uni-directional training, bi-directional inferencing".  Reviewers questions the extent of technical novelty and performance gaps, esp. on segmentation (10% worse than Mask2Former).  While authors explains the cause of this gap by pointing out that their model cannot leverage the transport capability of Stable Diffusion between noises and images for the transport between images and masks,  this inability does cast doubts on the flexibility and practicality of this framework.

Nevertheless, the AC recommends acceptance based on the consensus of an interesting novel approach to unify low-level and high-level vision tasks, with non-trivial technical solutions.  Please take all the comments and additional clarification/results in the rebuttal into account and improve the quality in the final version.